# Natural Populations from the *Phytophthora palustris* Complex Show a High Diversity and Abundance of ssRNA and dsRNA Viruses

**DOI:** 10.3390/jof8111118

**Published:** 2022-10-24

**Authors:** Leticia Botella, Marília Horta Jung, Michael Rost, Thomas Jung

**Affiliations:** 1Phytophthora Research Centre, Department of Forest Protection and Wildlife Management, Faculty of Forestry and Wood Technology, Mendel University in Brno, Zemědělská 3, 613 00 Brno, Czech Republic; 2Department of Genetics and Agrobiotechnology, Faculty of Agriculture and Technology, University of South Bohemia in České Budějovice, Na Sádkách 1780, 370 05 České Budějovice, Czech Republic

**Keywords:** RNA-sequencing, *Phytophthora*, mycovirus, oomycetes, virus evolution, virus ecology, natural habitat, multiple viral infections, virus reservoirs

## Abstract

We explored the virome of the “*Phytophthora palustris* complex”, a group of aquatic specialists geographically limited to Southeast and East Asia, the native origin of many destructive invasive forest *Phytophthora* spp. Based on high-throughput sequencing (RNAseq) of 112 isolates of “*P. palustris*” collected from rivers, mangroves, and ponds, and natural forests in subtropical and tropical areas in Indonesia, Taiwan, and Japan, 52 putative viruses were identified, which, to varying degrees, were phylogenetically related to the families *Botybirnaviridae*, *Narnaviridae*, *Tombusviridae*, and *Totiviridae*, and the order *Bunyavirales*. The prevalence of all viruses in their hosts was investigated and confirmed by RT-PCR. The rich virus composition, high abundance, and distribution discovered in our study indicate that viruses are naturally infecting taxa from the “*P. palustris* complex” in their natural niche, and that they are predominant members of the host cellular environment. Certain Indonesian localities are the viruses’ hotspots and particular “*P. palustris*” isolates show complex multiviral infections. This study defines the first bi-segmented bunya-like virus together with the first tombus-like and botybirna-like viruses in the genus *Phytophthora* and provides insights into the spread and evolution of RNA viruses in the natural populations of an oomycete species.

## 1. Introduction

Global trade has broken down the natural distribution ranges of species by enabling the long-distance movement of living organisms (including plant pests and pathogens) around the world and their establishment in new territories [1,2,3,4]. Alien invasive tree pathogens are recognized world-wide as a rising hazard to biodiversity and ecosystem functioning [2,5,6]. Therefore, the development of successful management strategies requires a profound understanding of the biology, epidemiology, and pathways of spread of the target organism [7]. Although challenging, identifying the original geographical and ecological niche of a given pathogen can not only help to predict the places where it may survive, but also identify natural antagonists, such as hyperparasites, which should be more frequent in the native range of their natural hosts than elsewhere [8,9].

A considerable proportion of known destructive alien forest and crop pathogens are oomycetes belonging to the genus *Phytophthora*, a group of eukaryotes phylogenetically related to brown algae, diatoms, and other Stramenopiles [10], which share much of their ecological niches, lifestyle, basic structural features, and virulence strategies with fungi [11]. Prominent examples of devastating emerging diseases include “chestnut ink disease” (caused by *Phytophthora cinnamomi* and *P. × cambivora*), “sudden oak death” in Europe, “sudden larch death” in North America (both caused by *P. ramorum*), and “Port-Orford-cedar (POC) root disease” (caused by *P. lateralis*) (reviewed in [6]). Recent surveys in natural ecosystems and population genetic studies demonstrated that many globally invasive *Phytophthora* pathogens are indigenous to Southeast and East Asia, indicating the region is an important center of origin of this genus [12,13,14,15,16,17]. Both Southeast and East Asia are hotspots of plant diversity due to their variety of geology, geomorphology, macroclimate, and orographic climates, their complex paleoclimatic history, the repeated immigration of plant species from northern latitudes, and the temporary connection of a multitude of islands to the mainland during glacial periods in the Pleistocene followed by interglacial separations [18]. Due to their coevolution with phylogenetically related tree species, *Phytophthora* species in yet unsurveyed regions, which do not cause serious diseases in their native ecological niches, could pose a risk to forests elsewhere [16]. One of these potential pathogens might be *Phytophthora* sp. Palustris, which was first detected alongside *P. cinnamomi* and *Phytophthora palmivora* in a Taiwanese subtropical lowland swamp forest on the Hengchun Peninsula in 2013 [14]. This taxon is only distantly related to its next relatives in *Phytophthora* Clade 9, including *P.* sp. 9 Hennops from river systems in South Africa, *P. virginiana*, and *P.* sp. lagoariana [14]. Between 2017 and 2019, isolates from several closely related *Phytophthora* taxa were obtained from rivers, mangroves, and ponds, and natural forests in subtropical and tropical areas in Indonesia and Japan. Collectively, the taxa from this “*P. palustris*” complex appear to be aquatic specialists without causing any apparent tree or plant disease [14]; T. Jung and M. Horta Jung, unpublished results. To facilitate readability, in the following text the “*P. palustris*” complex is referred to as “*P. palustris*”.

Virus research in forest sciences has been historically driven by the goal of understanding how viruses produce hypovirulence (decrease in sporulation and growth) on important tree pathogens (fungi and oomycetes) and how they could be efficiently applied as biological control agents (BCAs) [19,20]. However, the majority of fungal and oomycete viruses (at this moment, both known as mycoviruses) produce cryptic infections and do not seem to generate phenotypic alterations in their hosts (reviewed in [21]). How viruses influence their host behaviour and, in consequence, their host populations must be a way of adaptation to modulate their own transmission rates [22,23] and/or must derive from complex environment–host–mycovirus interactions [24], because viruses and their hosts are part of a holobiont in an ecosystem rather than living in isolation [23]. Thus, studying the populations of a tree pathogen and its viruses in undisturbed natural forests (endemic ecological niches) [19], where both are naturally embedded, could lead to a better understanding of the virus effect on host behaviour, virus–host coevolution patterns, and the impact of biotic and abiotic factors on the virus distribution and transmission [25,26]. Moreover, mycoviruses replicate only in the cytoplasm of their hosts. A majority of them do not have any extracellular stages and are transmitted inside cells through hyphal anastomosis and heterokaryosis (lateral or horizontal transmission), or via spores (vertical or serial transmission) [27,28]. Such mechanisms indicate that the evolution of mycoviruses may mirror that of their hosts [29]. In general, host diversification is the result of geographical separation and ecological adaptation, and viral communities can provide an insight into the history of dispersion, with a particular relevance to the establishment of new populations [30]. Mycoviruses can contribute to the identification of the invasion history of tree and plant pathogens. Cryphonectria hypovirus 1, which infects the chestnut blight fungus *C. parasitica*, was revealed to have been introduced in Europe along with its fungal host. It rapidly colonized the expanding host population [20]. For Ustilago maydis virus H1 (UmvH1), infecting the pathogen of maize *Ustilago maydis*, a similar scenario was detected, namely that the virus was spread together with the fungus from Mexico to the United States [30]. The conifer pathogen *Gremmeniella abietina* (biotype A) was introduced together with its gammapartitivirus from Europe to the United States [31]. In addition, Hymenoscyphus fraxineus mitovirus 1 (HfMV1) confirmed the hypothesis that only two (mitovirus-carrying) *H. fraxineus* individuals were brought into Europe from Japan [32].

An increasing number of viruses have lately been described from the genus *Phytophthora*. Single virus infections, but also, very often, multiple infections, seem to be common. Double-stranded (ds) RNA viruses related to the families *Totiviridae*, *Megabirnaviridae*, and the proposed “Fusagraviridae” and “Ustiviridae”; positive (+) single-stranded (ss) RNA viruses related to *Narnaviridae and Endornaviridae*; and negative (−) ssRNA viruses related to the order *Bunyavirales* have been described in *Phytophthora* species, including *P. infestans*, *P. cactorum*, *P. condilina*, *P. castaneae*, *P. ramorum*, and a species infecting *Asparagus officinalis* in Japan [33,34,35,36,37,38,39,40]. In addition, recent studies have demonstrated the endogenization of a giant virus in the genome of *P. parasitica* (syn. *P. nicotianae*) [41]. The latest advances in *Phytophthora* virus research show relatedness to viruses found in other oomycete genera, including *Halophytophthora*, *Plasmopara*, and *Pythium* [42,43,44,45,46,47], as well as to fungi dwelling in a variety of ecosystems, plants, algae, and aquatic and soil invertebrates [33,46]. Nonetheless, the catalogue of oomycete viromes remains largely incomplete and only a few oomycete species have been investigated yet. Studying metatranscriptomic and metagenomic data offers a unique opportunity to study the virus diversity and abundance of these organisms, which are ubiquitous in marine, freshwater, and terrestrial environments.

Our major goals were (i) to study the virus diversity and abundance in an endemic *Phytophthora* species complex without reported global distribution; and (ii) to investigate the potential for these viruses to spread by comparing the virus community structure of the different “*Phytopththora palustris*” populations. By answering these questions, we can gain general insights into the epidemiological relevance of *Phytophthora* viruses in forests.

## 2. Materials and Methods

### 2.1. “Phytophthora palustris” Sampling and Isolation

Randomly selected rivers and streams were sampled during several *Phytophthora* surveys in Taiwan, Japan, and the Kalimantan and Sumatra islands in Indonesia during the years 2013, 2018, and 2019 (Appendix A). Forest rivers, rivers outside of a forest, mangroves, forest swamp soils, nursery ditches, and ponds were sampled. Naturally fallen leaves floating on the waterbodies were collected and directly plated onto selective PARPNH-agar [14]. Soil sampling and isolation methodology using young *Fagaceae* leaves as baits was performed, according to Jung et al. [14].

The complete list of “*P. palustris*” isolates used for this study and details of their sampling localities, dates, and collectors are given in Appendix A.

### 2.2. RNA Extraction

The total RNA of 112 isolates of “*P. palustris*” was purified from approximately 100 mg of fresh mycelium using an RNAzol^®^ RT Column Kit [48] and treated with a TURBO DNA-free™ Kit (Thermo Fisher Scientific, Waltham, MA, USA). RNA quantity was checked in a Qubit^®^ 2.0 Fluorometer (Thermo Fisher Scientific, Waltham, MA, USA). RNA quality was checked by Tape Station 4200 (Agilent), resulting in an RNA integrity number (RIN) of at least 7. Ten pools of RNA were prepared according to the provenance of “*P. palustris*” isolates and RNA quality (Appendix A). Pool JP-TW contained RNA from seven Japanese and five Taiwanese isolates; both JP1 and JP2 contained nine RNAs from Japanese isolates; KA1, KA2, and KA3 enclosed, respectively, 12, 10, and 8 RNAs from Kalimantan isolates; and pools SU1, SU2, SU3, and SU4 enclosed, respectively, 12, 12, 14, and 14 RNAs from Sumatran isolates.

### 2.3. RNA Library Preparation and Sequencing

Approximately 1 μg of total RNA eluted in RNase-free water was sent to SEQme s.r.o (Dobris, Czech Republic) for RNA library construction and deep sequencing. The rRNA was depleted with an NEBNext rRNA Depletion Kit (Human/Mouse/Rat) and constructed with an NEBNext Ultra II Directional RNA Library Prep Kit for Illumina and NEBNext Multiplex Oligos for Illumina (Unique Dual Index Primer Pairs) (NEB, Ipswich, MA, USA). Library QC was assessed in an Agilent Bioanalyzer 2100 High sensitivity DNA Kit. A KAPA Library Quantification Kit for Illumina platform was used for absolute, qPCR-based quantification of the Illumina libraries flanked by the P5 and P7 flow cell oligo sequences. Libraries underwent paired-end (PE) (2 × 150 nt) sequencing on a NovaSeq6000 (DS-150) (Illumina, San Diego, CA, USA) using a NovaSeq S4 v1.5 reagent kit. An “in-lane” PhiX control spike was included in each lane of the flow cell.

### 2.4. De Novo Virus Assembly and Detection Workflow

Raw data were automatically processed by the BaseSpace cloud interface (Illumina) in default settings. The basecalling, adapter clipping, and quality filtering were carried out using Bcl2fastq v2.20.0.422 Conversion Software (Illumina). The quality of raw reads was checked using FastQC (v0.11.9) (https://www.bioinformatics.babraham.ac.uk/projects/fastqc/, accessed on 28 August 2022) and MultiQC (v1.9) (https://multiqc.info/, accessed on 28 August 2022). The raw data were cleaned from low-quality reads (quality Phred score cutoff: <20), and adapter sequences and very short sequences (≤25 bp) were removed for both reads before sequence pairing using Trim Galore (0.6.4_dev) (https://github.com/FelixKrueger/TrimGalore, accessed on 28 August 2022). The trimmed reads were aligned to host reference sequence using BWA mem (v0.7.17-r1188) (https://github.com/lh3/bwa, accessed on 28 August 2022) with default settings. Since the real host genome is not available, the genome *of Phytophthora parasitica* was randomly chosen and used as a reference, but two closer genomes became recently available (*Phytophthora quininea* strain Ex-type BL 54 and *Phytophthora macrochlamydospora* strain Ex-type BL). The unmapped reads were extracted and converted to fastq format using SAMTOOLS 1.7. (v1.7) (https://github.com/samtools/, accessed on 28 August 2022) and BEDTOOLS 1.7. (v2.29.0) (https://github.com/arq5x/bedtools2). These reads were used as input to the de novo assembly step using SPAdes genome assembler (v3.11.1) (https://github.com/kbaseapps/kb_SPAdes, accessed on 28 August 2022). The contigs for each sample were aligned using BLAST+ (v2.9.0+, BLASTn and BLAST) [49] to viral protein reference, non-redundant viral protein, host genome, mitochondrion, *Phytophthora* spp., rDNA, Boseq sp AS-1, bacterial genome, and cds sequences. All contigs with >79.9% similarity to host were eliminated.

Consequent analyses, including the determination of the final virus sequences, primer design, detection of open reading frames (ORF), protein translation, pairwise (pw) sequence comparison (PASC), and read number calculation, were performed using the platform Geneious Prime^®^ 2021.0.4. and 2022.0.1. For the calculation of the coverage depth, we used the following formula: (Total reads mapped to the final identified virus * average read length)/virus genome or contig length).

### 2.5. Genetic Variability Analyses and Conserved Domains

Pairwise identities of the nucleotide and amino acid sequences were obtained after aligning the viral nucleotide and amino acid sequences by MAFFT V1.4.0 and calculated in Geneious Prime^®^ 2021.0.4. In order to search for conserved domains within the putative viral proteins, the NCBI CDD-search tool was used (https://www.ncbi.nlm.nih.gov/Structure/cdd/wrpsb.cgi, accessed on 28 August 2022).

### 2.6. Phylogenetic Trees

Maximum likelihood (ML) phylogenetic trees were constructed using a rapid bootstrapping algorithm [50] in RAxML-HPC v.8 on XSEDE, conducted in CIPRES Science Gateway [51]. Tree search was enabled under the GAMMA model to avoid thorough optimization of the best scoring ML tree at the end of the run. The Jones–Taylor–Thornton (JTT) model was chosen as a substitution model for proteins. Bootstrapping was configured with the recommended parameters given by CIPRES Science Gateway. The resulting data were visualized using the software FIGTREE version 1.4.4.

### 2.7. Rapid Amplification of cDNA Ends (RACE) and Confirmation of Viruses’ Occurrence by Direct Reverse-Transcription Polymerase Chain Reaction (RT-PCR)

To confirm the length of the RNA 1 of botybirna-like virus 2 and RNA 1 and 2 of bunya-like virus 11 (see Results), we used the SMARTer RACE 5′/3′ KIT (TAKARABIO USA, Inc., Mountain View, CA, USA), as described in [46], and following the producer’s instructions. The occurrence of each identified virus was confirmed by direct reverse-transcription polymerase chain reaction (RT-PCR) with specific primers using total RNA as a template. The screening did not differentiate among variants, and the variant-level prevalence was not studied. A High-Capacity cDNA Reverse Transcription Kit (Applied Biosciences, Park Ave, NY, USA) was used for the cDNA synthesis. PCRs were performed with a Hot Start Taq 2× Master Mix (New England BioLabs, Ipswich, MA, USA) including 25 µL Master Mix, 1 µL of each primer (10 mM), and 4 µL of cDNA in a total volume of 50 µL. RT-PCR products were visualized using gel electrophoresis (120 V; 60 min). Analyzed fragments were separated on 1.5% agarose gel prepared with a TBE 1X buffer (Merck KGaA, Gernsheim, Germany) and stained by Ethidium bromide (SIGMA-Aldrich, Steinheim, Germany). PCR products showing the amplicons of expected length were purified and sequenced by GATC BioTech (Eurofins; Konstanz, Germany) by both directions with the primers used for PCR amplification. All the primers used for the partial amplification of the RNA-dependent RNA polymerase (RdRP) of each virus (Appendix A) were designed by Primer 3 2.3.7 under Geneious Prime^®^ 2020.0.4.

### 2.8. Graphs and Visualizations

Maps and graphs were created based on physically measured GPS coordinates using the R version 4.0.5 [52] programming environment using the special libraries ggplot2, sf, mapplots, ggspatial, scatterpie, and ggrepel.

## 3. Results

### 3.1. Virus Identification

A total of 10 RNA libraries containing 10–15 RNAs pooled from “*P. palustris*” isolates from Japan, Taiwan, and Indonesia (Kalimantan and Sumatra) were constructed and sequenced (Table 1 and Appendix A). After trimming and quality checking, ~5200 million reads were obtained (Appendix A). Further removal of reads mapped to the host genome yielded a total of 259,086,400 reads. The number of host-mapped reads was very variable among the different pools (Appendix A) but generally very high, always higher than 84%. The highest percentage was obtained in pools JP2, SU2, and SU4 (>98%) and the lowest in pool KA1 (84.41%). A total of 730,972 possible viral contigs (length > 0–25,000 bp) were obtained by de novo assembly of viral contigs (Appendix A). After being aligned using BLAST+ to all the databases mentioned in the Materials and Methods section, virus contigs were finally identified to represent 52 viruses related to five virus families and orders: *Totiviridae* and *Botybirnaviridae* with dsRNA genomes; *Bunyavirales* with (−)ssRNA genomes; and *Narnaviridae* and *Tombusviridae* as (+)ssRNA genomes (Table 1). No DNA viruses were identified applying our pipeline.

### 3.2. Virus Genomic Organization and Phylogenetic Relationships

#### 3.2.1. (−)ssRNA Viruses

Based on the BLASTX comparison, a total of 18 viral contigs with affinities with the L (large) segment of putative members of the order *Bunyavirales* were assembled and characterized in Sumatran and Kalimantan pools (SU1-4 and KA1-3) (Table 1). All of them enclosed a unique large open reading frame (ORF) encoding the RNA-dependent RNA polymerase (RdRP) (Figure 1a). In addition, one contig showed a high identity percentage with nucleocapsid proteins (NC) of the family *Phenuiviridae*. The 17 putative viruses were designated as Phytophthora palustris bunya-like viruses (PpaBLV) (Table 1). The PASC of the 17 bunya-like RdRP nucleotide (nt) and amino (aa) sequences showed an overall pw identity of 34 and 39.4%, respectively. When all the RdRP nt and aa sequences were compared, several contigs were seen to have a pw identity >90% (Appendix A); therefore, those sequences were designated as variants of the same putative virus. Thus, both PpaBLV 9 and 14 had three variants. In ICTV, there are no primary classification and delimitation criteria for genus and species in the order *Bunyavirales*, and PASC and phylogenetic analyses seem to be the main point of reference to name new bunyaviruses.

Conserved domains (CDD) of the Bunya_RdRP superfamily cl20265 were found in bunya-like virus contigs 1, 2, 3, 4, 5, 6, 8, 9 (variant 1), 10, 11, and 13 (Table 1). The amino acid alignment of the CDD regions (Appendix A) showed high similarities of the premotif A and motifs A (DxxxWx), B (XGxxNxxSS), C (SDD), D (KK), and E (ExxSx) with the rest of the bunyaviruses included in the alignment. Premotif A with the three basic residues inside (K, R, and R/K) and, downstream, the glutamic acid (E), were also identified. The conserved aa triplet TPD (threonine), typical of bunyaviruses, was also detected [53]. The aa sequence of the NC of the Phytophthora palustris bunya-like virus 11 encloses a conserved motif of pfam05733, the only member of the superfamily cl05345 with a significant e-value (2.08 × 10^−7^) (Table 1, Figure 1a). This family consists of several *Tenuivirus* and *Phlebovirus* nucleocapsid proteins (Appendix A, Table 1). In addition to the conserved domain of Bunya_RdRP, an L-protein N-terminus (and endonuclease domain) present in the N-terminus of many bunyavirus L proteins was found at nts 7912–8166.

On the basis of the RACE analysis, we determined the possible full viral genome sequence of PpaBLV11, which comprised RNA1 (6448 nt) and RNA2 (864 nt) (Figure 1a). The RNA2 sequence is shorter than the NGS contig (ca. 1 kb) because only 2 relatively short sequences of 3′ were obtained by the RACE analysis. Moreover, the coverage of the ends of the NGS-RNA2 contig was very low and had very low quality. The 5′ and 3′ termini of RNA1 and RNA2 had 9 and 8 conserved nts, respectively (Figure 1b). A total of 9 sequential nts at the 5′ and 3′ termini of RNA1 and 8 nts in RNA 2 were complementary to each other (Figure 1c). The comparison of the 5′ and 3′ terminal nucleotides of PpaBLV11 to those of other bunyaviruses showed a high identity. Higher variation appears to occur in the 3′ terminus of the RNA2, having a C, which is lacking in the rest of the 3′ termini of the compared bunyaviruses; similarly, no complementary G is found in the 5′ terminus of PpaBLV1 RNA1.

ORF1, the unique large ORF found in RNA1, encodes a putative protein of 2084 aa (codons) and a molecular weight of 241.396 kDa (p241), and 5′ and 3′ untranslated regions (UTRs) comprising 72 and 121 nt, respectively (Figure 1a). ORF2, the only ORF found in RNA2, encodes a putative protein of 230 aa and a molecular weight of 26.459 kDa (p26).

A comprehensive reconstruction of the phylogenetic relationships of “*P. palustris*” bunya-like viruses’ RdRP with other bunyaviruses deposited in the GenBank is shown in Figure 2a. The majority of “*P. palustris*” bunya-like viruses cluster together and seem to be closer to Phytophthora condilina negative-stranded RNA virus 4 (PcoNSRV4), Halophytophthora RNA virus 1 (HRV1), both viruses described in marine oomycetes collected in estuarine ecosystems in southern Portugal [29,42], and to Phytophthora *cactorum* virus 1 (PcBV2), described in several isolates of *P. cactorum* causing crown rot in strawberries in Finland [33]. PpaBLV5 appears jointly with Phytophthora *cactorum* bunya virus 1 [33] in a closely related cluster, which just hosts bunya-like viruses found in oomycetes. PpaBLV6 is the most distinct taxon, clustering with Phytophthora condilina negative-stranded RNA virus 6 (PcoNSRV6) [29]. These results indicate a close evolutionary relationship among bunya-like viruses, described in different species of oomycetes collected from different environments.

The NC of PpaBLV11 appears to be phylogenetically related to NCs of an insect virus (genus *Phasivirus*) and a plant virus, and not related to those of fungal phenuiviruses (Figure 2b).

#### 3.2.2. (+)ssRNA Viruses

Tombus-like virus. One viral contig of 4180 bp found in the Sumatran pool SU2 appeared to have phylogenetic affinities with unclassified members of the family *Tombusviridae*. Although the best hit subjects in the BLASTX comparison were sequences obtained in a metatranscriptomic study from soil dominated by stands of *Avena* species (Table 1), other related results were Leuven tombus-like virus 3, described from larvae of predatory wasps in Belgium (e-value 8 × 10^−49^; identity ~32%), and Serdyukov virus (e-value 5 × 10^−45^; identity ~36%), reported in the ticks of Antarctic penguins [54].

When using the standard translation code, the contig encodes two positive-sense ORFs (Figure 3a), and the ORF1 or 5′ end-proximal ORF encodes an unknown protein with no conserved domains detected. The ORF2 or 3′ end-proximal ORF encodes the RdRP. The conserved domain of the RT-like superfamily (cl2808) is located at nt positions 1797–3191 (e-value 2.11 × 10^−57^), particularly, the superfamily member pfam00998 (Viral RdRP_3) found in Hepatitis C virus and other plant viruses (Appendix A). A third antisense ORF (nts 635-63) is detected proximal to 5′ end and just separated by one proline (Pro) from the next ORF1. The resultant protein (191 aa) does not have any similar records when blasted to GenBank. This contig was named Phytophthora palustris tombus-like virus 1 (PpaTbLV1). The phylogenetic tree (Figure 3b) shows that PpaTbLV1 is more related to tombusviruses infecting insects rather than those infecting the obligate biotrophic oomycetes *Sclerophthora macrospora* [55] and *Plasmopara halstedii* [42].

Narna-like viruses. Virus contigs resembling members of the family *Narnaviridae* were abundant in all pools. After removing redundant contigs and selecting the longest ones, 10 viruses were determined. Eight of them had lengths longer than 2.5 kb and enclosed a complete single large ORF, likely representing nearly full-length virus genomes (Table 1; Figure 4a).

Two contigs contained incomplete ORFs and represented partial genomes of two potential virus isolates. No narnavirus conserved domains were detected in any of the putative virus contigs in the Conserved Domain Database (CDD) but the BLASTX search confirmed their similarity with unclassified narna-like viruses found in the fungal pathogen *Eryshiphe necator* and in invertebrates, and with Phytophthora infestans RNA virus 4 (Table 1). The alignment of the conserved motifs of RdRP of the narna-like viruses is shown in Appendix A. The overall PASC of the eight narna-like contig nt and RdRP aa sequences were 34.2 and 27.1%, respectively. Based on PASC in the nt and aa sequences (Appendix A) and the species demarcation criterion proposed in the ICTV for the family *Narnaviridae*, eight possible viruses designated as Phytophthora palustris narna-like viruses 1-8 (PpaNLV1-8) were represented in these 10 contigs. PpaNLV3 was represented by three variants as the pw identity of their aa sequences was higher than 90%.

The phylogenetic analyses, based on the aa sequences of their RdRP (Figure 4b), showed that “*P. palustris*” narna-like viruses are grouped in two different clusters. PpaNLV2, 3 (including all 3 variants), 4, 5, and 8 appeared in a detached group, maybe representing a different taxon within the “alphanarnavirus” clade, and closer to Saccharomycces 23S RNA narnavirus and Saccharomycces 20S RNA narnavirus. Within the clade, “betanarnaviruses” PpaNLV1 and 6 appear together with narnaviruses described in *Phytophthora* spp. and PpaNLV7 groups with two unclassified viruses described in an insect and in symptomatic grapevine tissue associated with the downy mildew *Plasmopara viticola*.

#### 3.2.3. dsRNA Viruses

Botybirna-like viruses. Four contigs >6 kb in length enclosing one large ORF resembling the RdRP segment of unclassified botybirnaviruses present in fungi and oomycetes (Figure 5a; Table 1) were found in two pools from Sumatra (SU1, SU3) and one from Kalimantan (KA3). No second segment was detected in our data. These viral contigs were named as Phytophthora palustris botybirna-like viruses (PpaBbLV). The PASC of the nt and aa sequences of the four contigs showed high identity between them (Appendix A), with particularly high identity levels (>92%) between the Sumatran contigs. Because the ICTV does not specify virus species demarcation criteria for this family, we classified the four sequences as belonging to two viruses: three contigs represent three Sumatran variants belonging to PpaBbL1 while one represents a Kalimantan botybirna-like virus. CDD belonging to the RT-like superfamily (cl2808) and, particularly, the superfamily member pfam02123 (RdRP_4, which includes RdRPs from *Luteovirus*, *Totivirus* and *Rotavirus*) were also found (Figure 5a and Appendix A; Table 1). The reconstruction of the phylogenetic relationships of PpaBbLV1 and 2 with other dsRNA viruses (Figure 5b) shows that PpaBbLV1 and 2 form a novel cluster separated from other botybirnaviruses found in fungi and oomycetes, suggesting an evolutionary diversification among the mycoviruses of this family.

Toti-like viruses. Virus contigs resembling members the family *Totiviridae* were also abundant in all pools. After removing redundant and shorter ones, 20 contigs were characterized and named Phytophthora palustris toti-like viruses (PpaTLV). Based on the PASC (Appendix A) and ICTV criterion for the species demarcation within the family *Totiviridae*, 14 *totiviruses* were described (Table 1), and those viral contigs with pw identity higher than 90% in both nt and aa sequences were considered variants of the same virus: PpaTLV1 (variants 1 and 2), PpaTLV2 (variants 1, 2, and 3), PpaTLV3 (variants 1, 2, and 3), PpaTLV4, PpaTLV5 (variants 1 and 2), PpaTLV6, 7, 8, 9, 10, 11, and PpaTLV12 (variants 1, 2, and 3). Most of the contigs consisted of ~5–6 kb sequences enclosing two ORFs. Only PpaTVL4 and 8 were shorter and contained one ORF. Conserved domains belonging to the superfamily member pfam02123 (RdRP_4) were found in the 3′-terminus proximal ORF in all the contigs (Figure 5a and Appendix A, Table 1). In addition, conserved domains of the totivirus coat superfamily, cl25797 and pfam05518, were found in the 5′-terminus proximal ORF of PpaTLV1-1, 1-2, 2-1, 2-2, 2-3, 5-1, 5-2, 6, 9, 10, and 11 (Appendix A, Table 1).

The phylogenetic relationships of PpaTLVs, reconstructed based on their aa RdRP sequences, show a high degree of evolutionary differentiation among them. Whilst PpaTLV1, 2, 5, 6, 10, and 11 appear to cluster with viruses belonging to the genus *Victorivirus* described in different fungi (Figure 5b), PpaTLV3 falls into the cluster with mycoviruses classified in the genus *Totivirus*. In a separated cluster, PpaTLV12 is phylogenetically closely related to viruses from *Plasmopara viticola*, *Pythium*, and *Phytophthora* spp. from diverse ecosystems, which form an oomycete-specific cluster related to Giardia canis virus (GCL). PpaTLV4, 7, and 8 cluster with unclassified toti-like viruses reported from fungi, oomycetes (including *P. condilina*), and diatoms.

### 3.3. Virus Abundance, Diversity, and Geographical Distribution

#### 3.3.1. Virus Occurrence Based on the RT-PCR Screening

All the viruses detected by RNA-Seq were confirmed to occur in 45 isolates of “*P. palustris*” from 25 Sumatra and Kalimantan localities (Figure 6) after performing ca. 1800 PCRs. No virus presence was confirmed by RT-PCR in Japanese and Taiwanese isolates.

In Sumatra, 9 (SU_F11, F42, R02, R04, R05, R09, R13, R16, R29) out of 30 sampling sites (30%) did not appear to host any of the screened viruses. However, in 21 of the locations (70%), at least one virus was detected. A total of 30 “*P. palustris*” isolates (58%) hosted at least one virus, and 15 of them showed viral coinfections or multiviral infections. In particular, isolate SU0376 appeared to be infected by seven distinct viruses while SU1474 was infected by six viruses (Figure 7a, Figure 8, and Figure 9). In Kalimantan, isolates from four out the five sites screened (KA_R03, R05, R06, R08) contained viruses. Out of 30 “*P. palustris*” isolates, 15 isolates hosted at least one virus (50%), and seven of them (47%) showed coinfections and multiviral infections. “*P. palustris*” isolates KA0119, KA0139, KA0146, and KA0156 hosted at least 11 viruses (Figure 7b, Figure 8, and Figure 9).

The virus richness appears to be greater in Sumatra (23 viruses) than in Kalimantan (17 viruses) (Figure 8 and Figure 9). Moreover, in our samples from Sumatra, the virus family composition was more diverse (Figure 7, Figure 8, and Figure 9), as a tombus-like virus (PpaTbV1) was found in two isolates collected from two Sumatran sites (SU_R06 and R07). Conversely, with 62 and 66 confirmed viruses, respectively, total virus abundance in “*P. palustris*” seems to be similar in Kalimantan and Sumatra. However, virus abundance in relation to the number of isolates tested from each of the two islands showed considerable differences: Kalimantan 63/30 = 2.1, Sumatra 66/52 = 1.27. Notably, with 14 different viruses, the Kalimantan site R06 hosted the highest virus diversity of the whole study.

Comparisons of the geographical distribution of “*P. palustris*” viruses in Sumatra and Kalimantan show that some viruses have a wider distribution than others (Figure 8 and Figure 9). Thus, five viruses, including three toti-like viruses (PpaTLV3, PpaTLV5, and PpaTLV12) and one narna-like virus (PpaNLV3), even occurred in “*P. palustris*” isolates from both Sumatra and Kalimantan. PpaTLV3 was found in seven isolates collected at five sites (three in Kalimantan and two in Sumatra). PpaTLV5 was the most widely distributed virus, infecting eight isolates from six different localities (two in Kalimantan and four in Sumatran). Finally, PpaTLV12 infected six isolates from five localities (one from Kalimantan and four in Sumatran). Within the same island, PpaTLV11 was present in 12 isolates from two Kalimantan localities, and PpaBLV9 infected 7 isolates obtained from seven Sumatran localities. Finally, the results show that, in general, toti-like viruses are more abundant and diverse in Kalimantan whereas bunya-like viruses are more prevalent and diverse in Sumatra.

According to the average pw identity percentages of the alignment of the amplicon sequences (variants) with the virus contig sequences obtained by NGS (Appendix A), some viruses also have more genetic stability than others, independently of their occurrence in two or more “*P. palustris*” isolates, i.e., the three variants of botybirna-like virus 1 (PpaBbLV1) are more diverse than the four variants of PpaBbLV2.

#### 3.3.2. Virus Read Abundance per Pool and Island

There is a large variation in the total viral read numbers between Japan, Taiwan, and Indonesia (Kalimantan and Sumatra) as well as between specific virus contigs. As shown in Appendix A, Sumatran libraries contained the highest percentage of viral reads, while Japanese and Taiwanese libraries contained the lowest. Likewise, the narna-like viruses constituted the highest read numbers (Table 1, Appendix A), with the PpaNLV 3-1, 3-2, 3-3, and 4 being clearly the most abundant viruses, occurring in libraries KA1, KA2, and SU1-SU3. Bunya-like viruses’ reads also appeared to be very abundant, in particular, the three variants of PpaBLV9 (found in libraries SU1–SU4). Interestingly, despite not having been confirmed by RT-PCR, a total of 608 reads were mapped to bunyavirus 8 (PpaBLV8) in all three libraries JP-TW, JP1, and JP2, while 68 reads were mapped to narna-like virus 8 (PpaNLV8). These reads scarcely covered these two virus sequences, repeatedly mapping short regions of those viruses. The abundance of toti-like and tombus-like viruses’ reads did not appear to be correlated with their distribution and richness. Compared to the high diversity and distribution of toti-like viruses, the number of reads for each contig was low, while the only tombus-like virus contig, which occurred mostly in SU2, had nearly 200,000 reads.

**Figure 8 jof-08-01118-f008:**
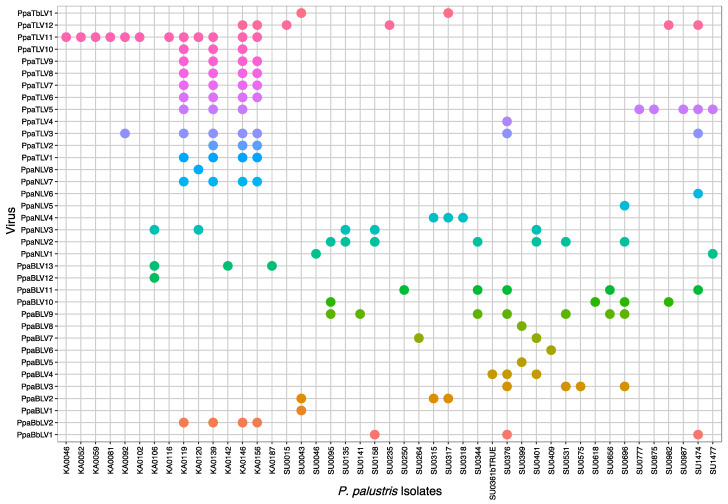
Plot representing the virus occurrence and diversity for each “*P. palustris*” isolate.

**Figure 9 jof-08-01118-f009:**
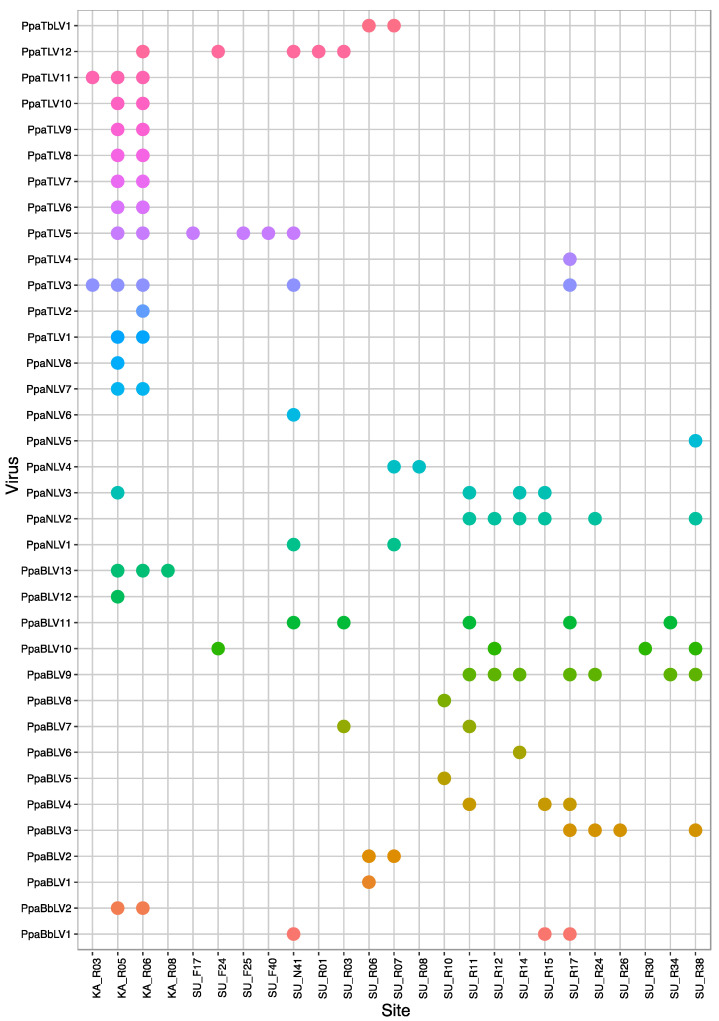
Plot representing the virus occurrence and diversity for each sampling site.

## 4. Discussion

The virus composition, abundance, and distribution discovered in our study indicate that viruses related to the families *Totiviridae*, *Tombusviridae*, *Narnaviridae*, and *Botybirnaviridae* and the order *Bunyavirales* are naturally infecting “*P. palustris*” isolates in their niches and must be readily carried by “*P. palustris*” propagules, revealing that they are primarily members of the host cellular environment.

### 4.1. Evolutionary Insights into Novel Taxa of (+)ssRNA Viruses

PpaTbLV1 is the first tombus-like virus described in a *Phytophthora* species, but not the first one in oomycetes. Four tombus-like viruses have been identified in *Plasmopara-viticola*-associated lesions on grapevines in Italy [43]. Plasmopara halstedii virus (PhV) and *Sclerophthora macrospora* virus A (SmV-A) have been detected in the downy mildew of sunflower *Plasmopara halstedii* [55] and the grass pathogen *Sclerophthora macrospora* [42], respectively, although PhV and SmV-A have been assigned to a new putative viral group between *Nodaviridae* and *Tombusviridae*. Based on the BLASTX and the phylogenetic analyses of this study (Figure 3b), PpaTbLV1 does not appear to be closely related to these viruses but to insect viruses and a virus found by a metagenomic approach in soils of wild oat (*Avena fatua*), an annual grass common in Mediterranean climates [56]. Interestingly, PpaTbLV1 is not grouped either with recognized plant *tombusviruses*, such as tomato bush stunt virus (TBSV) or with fungal tombus-like viruses (mycotombusviruses), such as Diaporthe RNA virus (DRV) [57]. However, PpaTbVLV1 genomic properties are closer to the latter, as shown by similar genome size (~4.2 kb), apparent absence of movement proteins typical of plant viruses, and the presence of two main ORFs encoding a hypothetical protein of unknown function and RdRP (Figure 3a). The RdRP is presumed to be expressed as a fusion product with the 5′-proximal protein via readthrough of the amber termination codon, UAG, of the upstream ORF [58]. However, PpaTbLV1 possesses GDD as the catalytic triplet in the motif-C of its RdRP (Appendix A), in contrast with a common feature in mycotombusviruses, which have GDN as the RdRP catalytic triplet [59]. Intriguingly, a smaller antisense ORF encoding an unknown protein is detected proximal to the 5′-terminus, which seems to be a unique property, not yet found in other tombus-like viruses. Thanks to high-throughput sequencing (HTS) studies, ambisense coding genomes emerge more and more commonly within RNA viruses and, particularly, in fungal viruses, such as ambiviruses [60] and “ambinarnaviruses” [61].

PpaTbLV1 was confirmed to be carried by two isolates of “*P. palustris*” collected in two Sumatran localities, within a distance of a few kilometers (Figure 8 and Figure 9, Appendix A). Hence, it seems possible that PpaTbV1 was transported by swimming zoospores and then exchanged via anastomosis between compatible individuals of “*P. palustris*”. Members of the “*P. palustris*” complex are aquatic oomycetes, and, interestingly, several reports have identified *tombusviruses* in association with rivers and lakes throughout the world [62,63]. TBSV and Carnation Italian ringspot virus (CIRV), have been isolated from waterbodies draining forest areas in Northrhine-Westfalia, Germany [64]. In addition, tombus-like viruses are also hosted by unicellular protists [65], they have been associated with the holobiont of freshwater shrimp [66], and have been discovered on marine and freshwater RNA viromes using metagenomics [67].

The family *Narnaviridae* is undergoing an intense reconstruction, as more and more new narna-like viruses with very diverse genomic properties have been discovered lately [58]. Until recently, it was generally understood that narnaviruses had monopartite (+) ssRNA genomes enclosing solely an essential RdRp gene encoded by a single ORF. However, it has been discovered that some narnaviruses have divided RdRPs (“splipalmiviruses”), i.e., viruses hosted by deep-sea fungi, such as *Aspergillus tennesseensis*. Others have bi- and multi-segmented genomes (reviewed in [58]) or possess long-reverse-frame ORFs, i.e., *Plasmopara-viticola*-associated narnaviruses [43]. Although we cannot rule out the possibility of segmented genomes, “*P. palustris*” narna-like viruses seem to be phylogenetically related to traditional monopartite positive-coding narnaviruses included in both “alphanarnaviruses” and “betanarnaviruses” clades (Figure 4b). Likewise, no reverse large ORFs were detected in any of the final contigs. The phylogenetic tree and the pairwise comparisons (Appendix A) show moderate levels of genetic variability. Whilst the narna-like viruses’ richness in “*P. palustris*” is remarkable (eight putative viruses are described), five of them are closely related and cluster together, suggesting they might have originated from the same “*P. palustris*” host and gradually incorporated genetic modifications over time [68]), probably during the replication process due to random mutations and genetic adaptations to a new geographical area [69,70,71]. A common ancestor might also explain the evolutionary relationship of PpaNLV1 with a narnavirus found in *Phytophthora castaneae* in Vietnam [40], and PpaNLV6 with a narnavirus found in *P. infestans* [34]. Narnaviruses seem to be effectively transmitted throughout “*P. palustris*” populations and are widespread in both Sumatra and Kalimantan. A certain relation between the high narnavirus abundance and read number obtained, in particular, for PpaNLV2, 3, and 4 (Table 1, Appendix A), agrees with the frequent occurrence of these viruses on both islands.

### 4.2. Discovery of New dsRNA Viruses and High Abundance of Toti-Like Viruses

Kalimantan and Sumatran isolates of “*P. palustris*” have been found to host two putative botybirna-like viruses (PpaBbLV1 and 2). While genetically highly similar (Appendix A), each PpaBbLV occurs in several isolates and more than one site but on separate islands, PpaBbLV1 in Sumatra and PpaBbLV2 in Kalimantan (Figure 8 and Figure 9). This constitutes the first discovery of a putative botybirnavirus in a *Phytophthora* species. Previously, two botybirnaviruses were identified in grapevine lesions associated with *Plasmopara viticola* in Spain and Italy [43], and in several fungal plant pathogens, including *Botrytis porri* [72] and *Sclerotinia sclerotiorum* [73]. Botybirnaviruses have a bipartite genome, with two types of structural proteins (p85/80, p70) and an RdRP. However, only the segment enclosing the RdRP was identified in our study. Both PpaBbLV 1 and 2 appear to be phylogenetically distinct from other mycobotybirnaviruses, representing a potential new taxon. Moreover, aside from motifs I and II, the rest of the RdRP domains closely resemble those of totiviruses (Appendix A).

“*P. palustris*” isolates are mainly infected by totivirus-related viruses. They are widespread in both Kalimantan and Sumatra (Figure 8 and Figure 9) and novel *Bona-fide* totiviruses, victoriviruses, and toti-like viruses show paraphyletic relationships across the phylogenetic tree and considerable genetic diversification (Figure 5b). Some viruses reside in an oomycete-specific cluster while others cluster with unclassified toti-like viruses reported from fungi, oomycetes and diatoms dwelling in a wide range of ecosystems. This result may suggest long-term coevolution between “*P. palustris*” and its totivirus-like viruses.

### 4.3. Persistence of Bunya-Like Viruses and the First Oomycete Bunyavirus with NC Protein

Similar to recent studies on marine oomycetes [33,46], high variability and abundance of the L segment (RNA 1) of putative bunyaviruses were also found in “*P. palustris*”, most of them phylogenetically closely related but with an overall high pairwise genetic variability that allows the description of 13 viruses with different variants. Such virus richness can be the consequence of mutation, recombination and reassortment during the course of evolution. Viral reassortment seems to be a powerful mechanism underlying the evolution of the *Bunyavirales* order, and it has been pointed out that most bunyaviruses described so far are actually reassortants of extant or extinct viruses [74]. Alternatively, it cannot be ruled out that the *P. palustris* bunya-like contig sequences would not represent true biological entities. Since bunyaviruses might be reassortants, which may sometimes be seen in their genomes as having highly conserved sequence stretches alternating with highly variable ones, putatively resulting in chimeric contigs in the de novo assembly when they are found abundantly in NGS-pools (e.g., [37]). The phylogenetic analysis (Figure 2a) shows the increasingly complex taxonomy of mycobunyaviruses with strong cluster support.

PpaBLV11 is the first bunya-like virus with bipartite genome reported in *Phytophthora*. It is related to phleboviruses, which seem to have a tripartite genome (three RNA segments). In this study, we only found two RNAs, RNA1 and RNA2, respectively, corresponding to the L and S segments of phleboviruses. Neither the M segment nor M segment-like contigs were detected during the NGS analysis. If PpaBLV11 has a tripartite genome, the M segment may not be considered essential. This is in accordance with the most similar virus, tulip streak virus (TuSV) [75] and fungal phleboviruses, as Lentinula edodes negative-strand RNA virus 2-HG3 (LeNSRV2-HG3) [76]. However, similar to the rest of the *Phytophthora* bunyaviruses, which seem to have monopartite genomes, it is more plausible that the genome description of these viruses is incomplete. The putative NC and other non-structural (Ns) associated proteins are likely not conserved enough to be detected by homology.

### 4.4. Virus Population Structure and Transmission Efficiency

Our study indicates that “*P. palustris*” viral populations adhere to a certain structure. Although the virus family composition is similar on both Kalimantan and Sumatra islands, the virus structure differs and appears to be specific to each island. In Kalimantan, toti-like viruses predominate, whereas bunya-like viruses are more prevalent in Sumatra. Furthermore, the absence of viruses in Japanese and Taiwanese “*P. palustris*” isolates suggests a degree of divergence regarding Indonesian members of the “*P. palustris*” complex and their viruses. However, it cannot be ruled out entirely that Japanese and Taiwanese isolates also host viruses. In fact, various viral reads have been mapped to PpaNV6 and PpaBLV8 in the RNA pools JP1, JP2, and TW-JP (Appendix A). If there are any viruses, they are not those found in Kalimantan and Sumatra. Since mycoviruses do not have an extracellular stage, they strongly depend on their host for dispersal, and an analogous evolutionary scenario is expected for both the host (“*P. palustris*”) and the hyperparasite (mycoviruses) [26]. However, due to geographical isolation, local adaptation, and drift, “*P. palustris*” seems to have evolved into new lineages, and the species *P.* sp. *palustris* in Japan and Taiwan is very different from all Indonesian taxa, indicating they have been separated from each other millions of years ago (T. Jung and M. Horta Jung, unpublished data), similarly to evolutionary divergent lineages of *P. ramorum* [16] and *P. lateralis* [13]. As a consequence of speciation processes in “*P. palustris*” hosts, the possibilities of the associated mycoviruses to coevolve and be always co-introduced with their host would significantly decrease. Multiple introduction routes do not necessarily imply that hyperparasites are always introduced along all of them [77]. Alternatively, the absence of viruses in these pools could be also due to the limited availability of similar viral sequences in GenBank; hence, a data re-analysis is highly recommended in a few years.

Our results suggest reasonable efficient viral transmission across Indonesian “*P. palustris*” populations. However, some “*P. palustris*” viruses are more widespread than others, merging in distinct Indonesian localities of the same island and even on different islands (PpaNLV3, and PpaTLV3, 4, and 12). Most likely these viruses evolved towards optimal virus–host interactions that enabled them to replicate inside their “*P. palustris*” hosts and move more efficiently through their host populations [77]. Members of the “*P. palustris*” complex are aquatic *Phytophthora* species and, hence, able to produce large amounts of motile chemotactic zoospores [78], which in waterbodies can be transported passively over long distances. However, “*P. palustris*” shares its aquatic habitat with nematodes, algae, crustaceans, and mollusks [79], with which they undoubtedly interact, and which might eventually serve as transport means of potential virus-hosting oomycete individuals. On the contrary, the lower occurrence of other viruses might be related to the lower efficiency of interspecies virus transmission, because five species of the “*P. palustris*” complex occur in Sumatra and Kalimantan (T. Jung and M. Horta Jung, unpublished data). This study did not observe a virus/“*P. palustris*” species correlation but it should be further investigated. Previous studies concerning virus transmission in *P. infestans* demonstrated 100% inheritance of Phytophthora infestans RNA virus 3 (PiRV3) in individual zoospores [80]. Phytophthora infestans RNA virus 2 (PiRV2) was readily transmitted by hyphal anastomosis, and by asexual reproduction through sporangia [80]. However, attempts to transfer PiRV-2 into apparently vegetatively incompatible *P. infestans* isolates failed.

### 4.5. Multiple Viral Infections at Isolate and Site Level

Particular Indonesian localities and “*P. palustris*” isolates appear to be big reservoirs of viruses (Figure 7, Figure 8 and Figure 9). From an ecological perspective, in those localities with higher virus diversity and abundance, the number of contact events between and among host individuals must be higher [26]. Thus, the probability of a virus-hosting individual having contact with another compatible virus-hosting “*P. palustris*” isolate increases with host population density. From an individual perspective, multiple viral infections are, indeed, known to occur in oomycetes in nature [23,37,40,43,46]. However, the interactions between different viruses and potential limits for the accumulation of multiple viruses in a single host remain unclear. In a native environment, long-term co-existence between virus and host should lead to a non-lethal equilibrium. Thus, the cytoplasmic exchange that takes place during fungal (and oomycete) growth promotes the accumulation of multiple viruses within a hypha, colony, or hyphal network [81]. Unlike fungi, oomycete hyphae lack septa, potentially facilitating the accumulation and exchange of viruses within and between compatible individuals. In addition, many *Phytophthora* species, in particular those thriving in aquatic environments, are prompted to interspecific hybridizations, which play a major role in speciation and species radiations in diverse natural ecosystems [82,83,84]. Interspecific hybridizations might enable virus transmission between different *Phytophthora* species.

## Figures and Tables

**Figure 1 jof-08-01118-f001:**
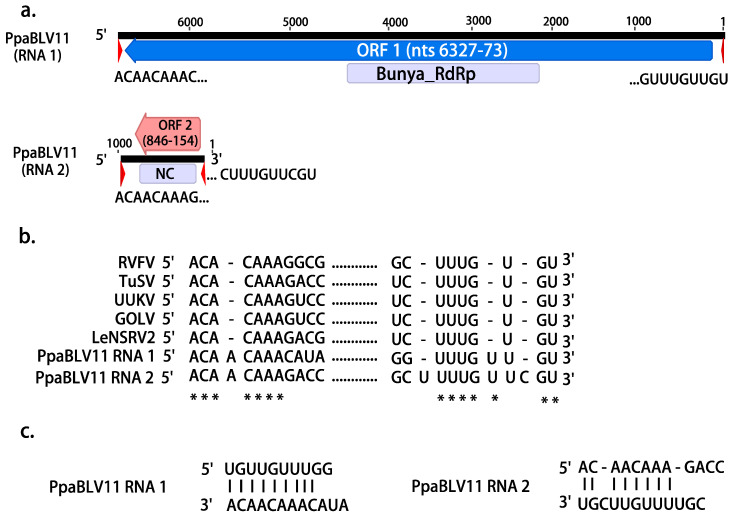
(**a**). Graphical representation of the L segment (RNA 1) and S segment (RNA 2) of PpaBLV11. (**b**). Alignment of 5′ and 3′ terminal sequences of PpaBLV11 RNA1 and RNA 2 with RNA1 or L segment sequences from the following viruses: Rift Valley fever virus (RVFV, DDBJ/EMBL/GenBank accession number: X56464), tulip streak virus (TuSV, LC571987), Uukuniemi virus (UUKV, D10759), Gouleako virus (GOLV, HQ541738), Lentinula edodes negative-strand RNA virus 2-HG3 (LeNSRV2, LC466007). Asterisks indicate that the nucleotides are 100% identical in all the viruses. (**c**). Complementary structure between the 3′ and 5′ termini in the putative PpaBLV11 genome.

**Figure 2 jof-08-01118-f002:**
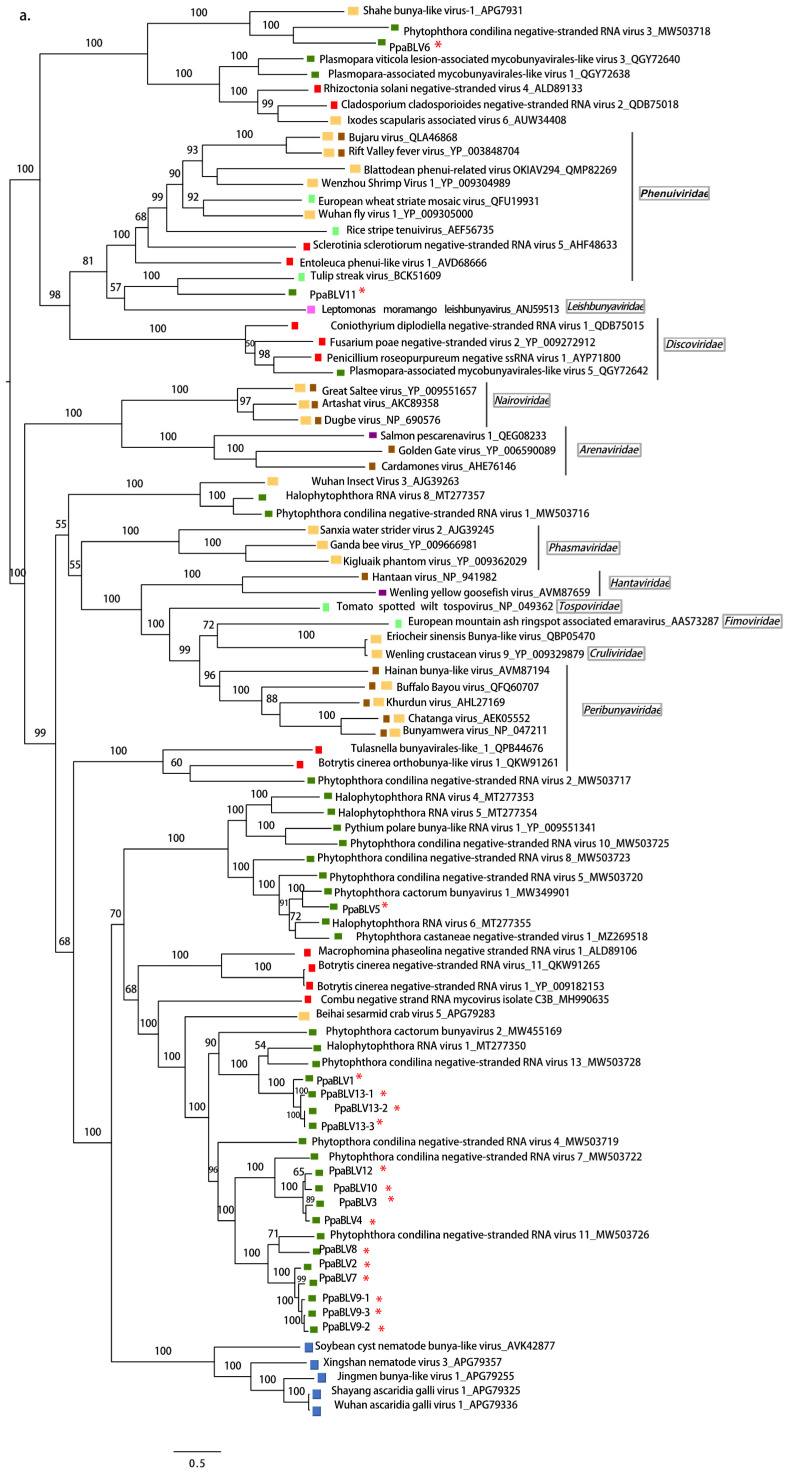
(**a**) Maximum likelihood tree (RAxML) depicting the phylogenic relationship of the predicted RdRP of Phytophthora palustris bunya-like viruses with other complete RdRP belonging to related viruses from the order *Bunyavirales*. (**b**) RAxML tree with the NC of PpaBLV11 and other bunyaviruses of the family *Phenuiviridae*. Nodes are labeled with bootstrap support values ≥50%. Branch lengths are scaled to the expected underlying number of amino acid substitutions per site. Phytophthora palustris bunya-like viruses 1–13 (and variants) are represented by their abbreviated names (PpaBLV1-13) and indicated with a red asterisk (*). Family classification and the corresponding GenBank accession numbers are shown next to the virus names. Colorful squares represent the virus host kingdom or phylum: 

 Fungi, 
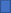
 Nematoda, 
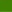
 Oomycota, 
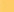
 Arthropoda, 
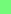
 Plants, 

 Mammalia, 
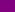
 (Fishes) Chordata, 
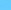
 Ochrophyta (Heterokonta), 
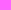
 Excavata. Scale bars represent expected changes per site per branch.

**Figure 3 jof-08-01118-f003:**
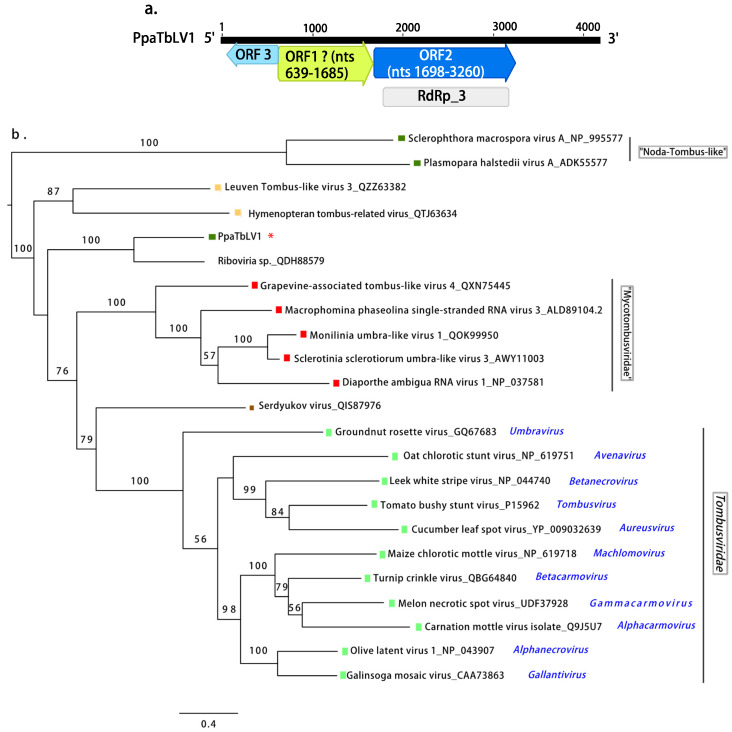
(**a**). Graphical representation of the tombus-like virus contig (PpaTbLV1). (**b**). Phylogenetic analysis (RAxML) based on the predicted RdRP of Phytophthora palustris tombus-like virus 1, abbreviated and indicated by a red asterisk (*), with other complete unclassified tombus-like viruses and members of the family *Tombusviridae.* Nodes are labeled with bootstrap support values ≥50%. Branch lengths are scaled to the expected underlying number of amino acid substitutions per site. Tree is rooted in the midpoint. Family classification and the corresponding pBLAST accession numbers are shown next to the virus names. Colorful squares represent the virus host kingdom or phylum: 

 Fungi, 
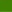
 Oomycota, 
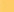
 Arthropoda, 
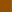
 Animalia, 
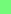
 Plants. Scale bar = 0.4 expected changes per site per branch.

**Figure 4 jof-08-01118-f004:**
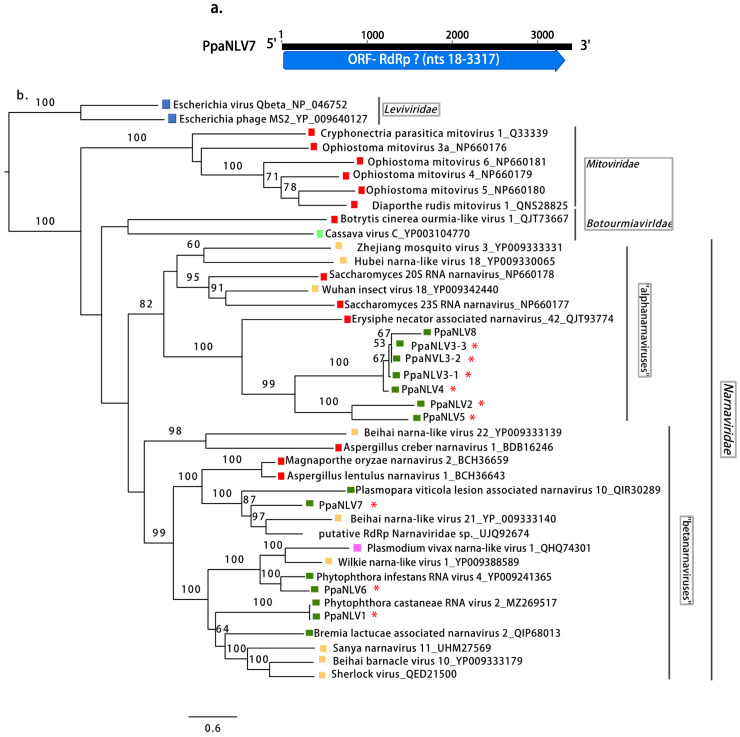
(**a**). Graphical representation of a narna-like (PpaNLV7) virus contig. (**b**). Phylogenetic analysis (RAxML) based on the predicted RdRP of *Phytophthora palustris* (+)ssRNA viruses with other complete classified and unclassified members of the families *Mitoviridae, Narnaviridae, Botourmiaviridae, Leviviridae*. Nodes are labeled with bootstrap support values ≥50%. Branch lengths are scaled to the expected underlying number of amino acid substitutions per site. Tree is rooted in the midpoint. Phytophthora palustris narna-like viruses are abbreviated PpaNLV1-8, and indicated with a red asterisk (*). Family classification and the corresponding pBLAST accession numbers are shown next to the virus names. Colorful squares represent the virus host kingdom or phylum: 

 Fungi, 
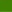
 Oomycota, 
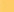
 Arthropoda, 
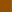
 Animalia, 
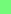
 Plants, 
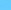
 Ochrophyta (Heterokonta), 
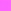
 Apicomplexa (Protists), 

 Bacteria. Scale bar = 0.6 expected changes per site per branch.

**Figure 5 jof-08-01118-f005:**
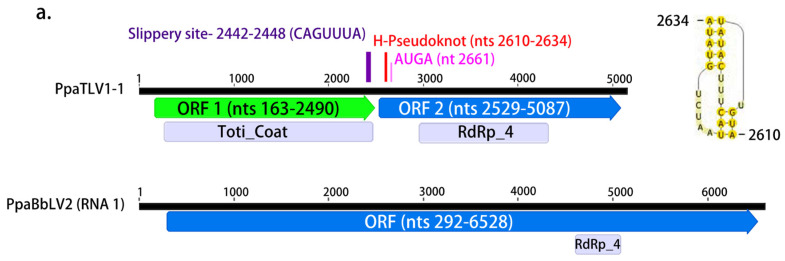
(**a**). Graphical representation of the genomes of one botybirna-like virus (PpaBbLV2) and one toti-like virus (PpaTLV1-1). (**b**). Phylogenetic analysis (RAxML) based on the predicted RdRP of the dsRNA viruses discovered in *Phytophthora palustris* with other classified and unclassified members of the families *Totiviridae* and *Botybirnaviridae*. Nodes are labeled with bootstrap support values ≥50%. Branch lengths are scaled to the expected underlying number of amino acid substitutions per site. Tree is unrooted and branches are shown in decreasing order. Variants of Phytophthora palustris toti-like and botybirna-like viruses are abbreviated PpaTLV1-12 and PpaBbLV1 and 2, respectively, and indicated with a red asterisk (*). Family classification and the corresponding pBLAST accession numbers are shown next to the virus names. Colorful squares represent the virus host kingdom or phylum: 

 Fungi, 
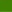
 Oomycota, 
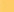
 Arthropoda, 
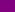
 Chordata, 
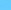
 Ochrophyta (Heterokonta), 
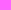
 Excavata. Scale bar = 0.6 expected changes per site per branch.

**Figure 6 jof-08-01118-f006:**
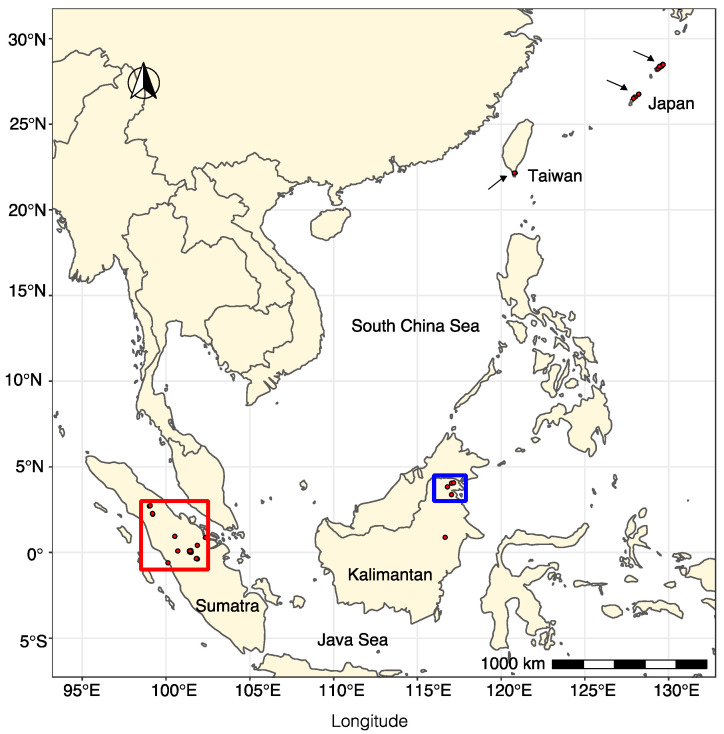
Map of Southeast and East Asia showing the sampling locations in Indonesia, Taiwan, and the Japanese Okinawa and Amami islands (indicated by red dots and arrows) where “*P. palustris”* isolates were collected.

**Figure 7 jof-08-01118-f007:**
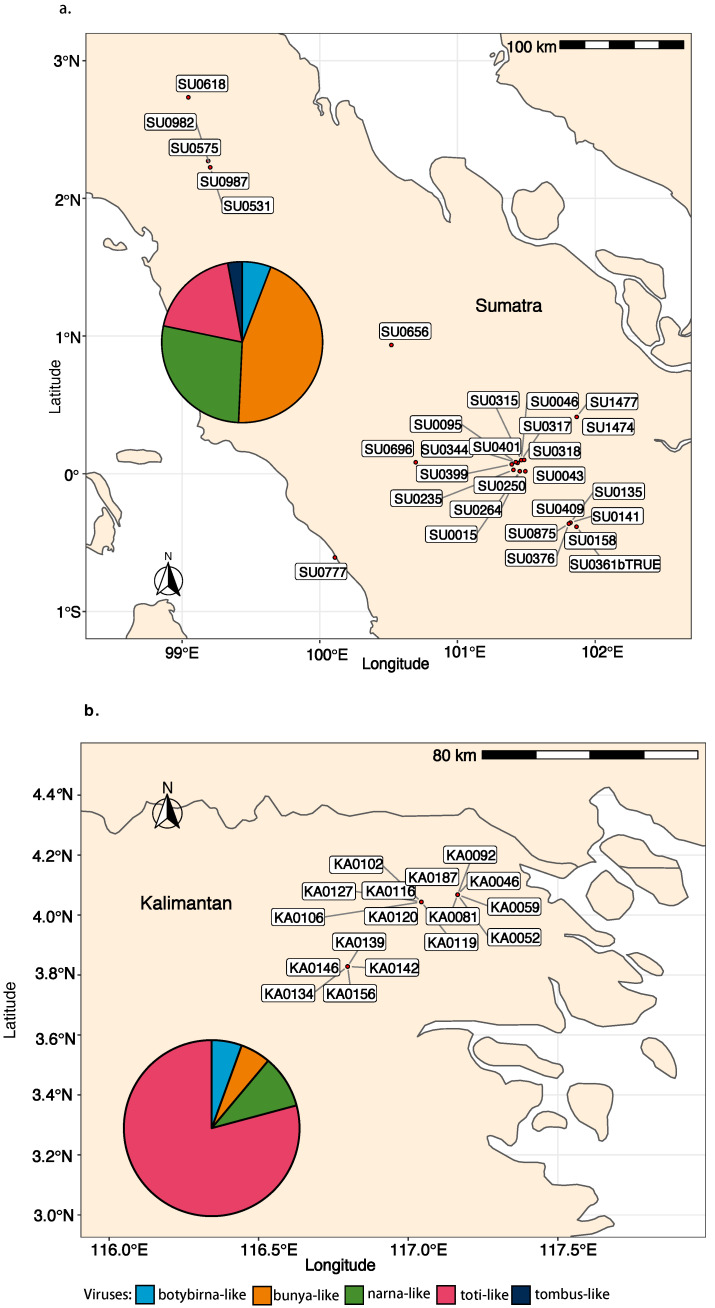
(**a**). Map showing the sites of the virus-hosting “*P. palustris*” isolates in Sumatra. (**b**). Map showing the sites of the virus-hosting “*P. palustris*” isolates in Kalimantan. The pie charts illustrate the relative frequency of each virus family/order in Sumatra and Kalimantan.

**Table 1 jof-08-01118-t001:** GenBank most similar viruses to the putative “*Phytophthora palustris*” viruses based on BLASTX search and parameters of their genome organization.

Acronym *	N ^N^	Genome	L ^L^ (bp)	Most Similar Virus in GenBank	E-Value	I (%)	QC (%)	CDD	CDD Position ^P^	R (bp) ^R^
PpaBbLV1-1	OL795338	dsRNA	6680	Plasmopara viticola lesion associated botybirnavirus 1	8.00 × 10^−27^	28.53	16	RdRp_4	4815–5834	2729
PpaBbLV1-2	OL795339		6680	Sclerotinia sclerotiorum botybirnavirus 3	3.00 × 10^−31^	27.94	18	RdRp_4	4821–5834	1831
PpaBbLV1-3	OL795340		6581	Plasmopara viticola lesion associated botybirnavirus 1	3.00 × 10^−32^	29.59	17	RdRp_4	4796–5809	6030
PpaBbLV2	OL795341		6514	Plasmopara viticola lesion associated botybirnavirus 1	4.00 × 10^−32^	29.13	18	RdRp_4	4669–5808	2804
PpaBLV1	OL795342	(−)ssRNA	9042	Phytophthora condilina negative-stranded RNA virus 13	0.0	48.86	98	Bunya_RdRp	4123–5085	173,414
								N terminus	7912–8166	7720
PpaBLV2	OL795343		9024	Phytophthora condilina negative-stranded RNA virus 11	0.0	56.29	97	Bunya_RdRp	4437–5114	114,896
PpaBLV3	OL795344		9303	Phytophthora condilina negative-stranded RNA virus 7	0.0	61.99	98	Bunya_RdRp	4338–5111	72,926
PpaBLV4	OL795345		9302	Phytophthora condilina negative-stranded RNA virus 7	0.0	56.27	98	Bunya_RdRp	4338–5828	43,584
PpaBLV5	OL795346		8362	Phytophthora cactorum bunyavirus 1	0.0	67.27	97	Bunya_RdRp	4082–5494	62,939
PpaBLV6	OL795347		6396	Phytophthora condilina negative-stranded RNA virus 3	0.0	33.97	84	Bunya_RdRp	2305–3342	10,639
PpaBLV7	OL795348		9036	Phytophthora condilina negative-stranded RNA virus 11	0.0	51.87	97	Bunya_RdRp	n.d.	47,152
PpaBLV8	OL795349		9737	Phytophthora condilina negative-stranded RNA virus 11	0.0	59.64	85	Bunya_RdRp	4091–5116	16,049
PpaBLV9-1	OL795350		9016	Phytophthora condilina negative-stranded RNA virus 11	0.0	52.06	98	Bunya_RdRp	4434–5045	59,989
PpaBLV9-2	OL795351		9047	Phytophthora condilina negative-stranded RNA virus 11	0.0	52.46	97	Bunya_RdRp	n.d.	271,404
PpaBLV9-3	OL795352		9015	Phytophthora condilina negative-stranded RNA virus 11	0.0	52.29	98	Bunya_RdRp	n.d.	93,848
PpaBLV10	OL795353		9417	Phytophthora condilina negative-stranded RNA virus 7	0.0	57.27	98	Bunya_RdRp	4451–5224	146,562
PpaBLV11	OL795355		6448	Tulip streak virus	1.00 × 10^−133^	27.69	67	Bunya_RdRp	2443–4482	223,284
	OL795354		878	Wuhan Fly virus 1	1.00 × 10^−06^	28.29	54	Tenui-Phenui NC	210–806	154,294
PpaBLV12	OL795357		9386	Phytophthora condilina negative-stranded RNA virus 7	0.0	55.64	99	Bunya_RdRp	n.d.	29,824
PpaBLV13-1	OL795358		8997	Phytophthora condilina negative-stranded RNA virus 13	0.0	50.63	95	Bunya_RdRp	3982–5055	14,451
PpaBLV13-2	OL795359		8997	Phytophthora condilina negative-stranded RNA virus 13	0.0	50.58	95	Bunya_RdRp	3982–5055	10,073
PpaBLV13-3	OL795360		8995	Phytophthora condilina negative-stranded RNA virus 13	0.0	50.54	95	Bunya_RdRp	3980–5053	9885
PpaTbLV1	OL795371	(+)ssRNA	4324	RdRp [Riboviria sp.] QDH88579	1.00 × 10^−139^	52.09	33	RdRp_3	2042–2845	173,414
								RNA_dep_RNAP	2060–2854	
PpaNV1	OL795361	(+)ssRNA	2818	Phytophthora castaneae RNA virus 2	0.0	97.01	90	RdRp	n.d.	135
PpaNV2	OL795362		2545	Erysiphe-necator-associated narnavirus 42	3.00 × 10^−31^	33.33	37	RdRp		243,122
PpaNV3-1	OL795363		2725	Erysiphe-necator-associated narnavirus 42	3.00 × 10^−21^	30.48	38	RdRp		138404
PpaNV3-2	OL795364		2719	Erysiphe-necator-associated narnavirus 42	3.00 × 10^−22^	31.02	39	RdRp		206,048
PpaNV3-3	OL795365		2736	Erysiphe-necator-associated narnavirus 42	2.00 × 10^−22^	31.28	38	RdRp		266,624
PpaNV4	OL795366		2761	Erysiphe-necator-associated narnavirus 42	1.00 × 10^−19^	33.33	27	RdRp		916,928
PpaNV5	OL795367		2576	Erysiphe-necator-associated narnavirus 42	1.00 × 10^−31^	37.34	28	RdRp		104,172
PpaNV6 ^U^	OL795370		1344	Phytophthora infestans RNA virus 4	4.00 × 10^−149^	59.45	97	RdRp		162
PpaNV7	OL795368		3381	Beihai narna-like virus 21	0.0	37.63	91	RdRp		125,312
PpaNV8 ^U^	OL795369		916	Erysiphe-necator-associated narnavirus 42	1.00 × 10^−11^	51.19	27	RdRp		46,432
PpaTLV1-1	OL795372	dsRNA	5134	Drosophila-associated totivirus 2	6.00 × 10^−171^	38.63	46	RdRp_4	2958–4325	5030
								Totivirus_coat	283–2487	
PpaTLV1-2	OL795373		5199	Drosophila-associated totivirus 2	8.00 × 10^−171^	38.63	46	RdRp_4	3023–4390	5464
								Totivirus_coat	348–2552	
PpaTLV2-1	OL795374		5183	Red algae totivirus 1	2.00 × 10^−162^	37.56	47	RdRp_4	3026–4381	4044
								Totivirus_coat	363–2624	
PpaTLV2-2	OL795375		5183	Red algae totivirus 1	5.00 × 10^−152^	38.31	47	RdRp_4	3026–4381	2962
								Totivirus_coat	363–2624	
PpaTLV2-3	OL795376		5183	Red algae totivirus 1	1.00 × 10^−151^	38.23	47	RdRp_4	3026–4381	2504
								Totivirus_coat	363–2624	
PpaTLV3-1	OL795377		4491	Conidiobolus heterosporus totivirus 1	0.0	42.03	49	RdRp_4	2548–3822	7076
PpaTLV3-2	OL795378		4491	Wuhan insect virus 26	0.0	41.34	49	RdRp_4	2623–3822	2174
PpaTLV3-3	OL795379		4495	Wuhan insect virus 26	0.0	41.03	49	RdRp_4	2627–3826	3750
PpaTLV4 ^U^	OL795380		1553	Diatom-colony-associated dsRNA virus 17 genome type B	6.00 × 10^−28^	32.59	57	RdRp_4	41–784	9026
PpaTLV5-1	OL795381		5188	Diatom-colony-associated dsRNA virus 11	8.00 × 10^−149^	38.41	47	RdRp_4	2969–4375	7926
								Totivirus_coat	486–2672	1546
PpaTLV5-2	OL795382		5199	Diatom-colony-associated dsRNA virus 7	5.00 × 10^−148^	39.47	43	RdRp_4	3019–4386	
								Totivirus_coat	497–2554	8402
PpaTLV6	OL795383		5312	Diatom-colony-associated dsRNA virus 11	1.00 × 10^−94^	35.22	37	RdRp_4	3118–4497	
								Totivirus_coat	389–2566	17,680
PpaTLV7	OL795384		5703	Diatom-colony-associated dsRNA virus 17 genome type B	2.00 × 10^−36^	27.67	27	RdRp_4	3663–4625	
PpaTLV8 ^U^	OL795385		2964	Diatom-colony-associated dsRNA virus 17 genome type A	4.00 × 10^−35^	27.29		RdRp_4	1216–2178	4168
PpaTLV9	OL795386		5268	Diatom-colony-associated dsRNA virus 11	0.0	48.39	52	RdRp_4	3075–4457	12,384
								Totivirus_coat	439–2550	
PpaTLV10	OL795387		5205	Red algae totivirus 1	3.00 × 10^−132^	36.31	41	RdRp_4	2979–4376	13,180
								Totivirus_coat	313–2355	
PpaTLV11	OL795388		5408	Red algae totivirus 1	6.00 × 10^−165^	40.69	39	RdRp_4	3219–4583	18,930
								Totivirus_coat	403–2424	
PpaTLV12-1	OL795389		4825	Plasmopara viticola lesion associated totivirus-like 1	0.0	46.10	44	RdRp_4	3030–3737	1786
PpaTLV12-2	OL795390		5833	Plasmopara viticola lesion associated totivirus-like 1	6.00 × 10^−179^	45.96	36	RdRp_4	4034–4741	13,412
PpaTLV12-3	OL795391		5833	Plasmopara viticola lesion associated totivirus-like 1	8.00 × 10^−178^	45.54	36	RdRp_4	4046–4753	15,479

n.d., not detected by CDD-search in https://www.ncbi.nlm.nih.gov/Structure/cdd/wrpsb.cgi, accessed on 28 August 2022. QC, query coverage. I, identity. ^P^ CDD position in 5′-3′ nucleotide sequence. ^R^ Total reads mapped to the final virus sequence. ^N^ GenBank accession numbers. ^L^ Final virus sequence length. ^U^ Partial segment and/or coding region. * PpaBbLV, Phytophthora palustris botybirna-like viruses; PpaTbLV, Phytophthora palustris tombus-like virus; PpaNLV, Phytophthora palustris narna-like viruses; PpaTLV, Phytophthora palustris toti-like viruses; PpaBLV, Phytophthora palustris bunya-like viruses. R Total number of reads mapped to each viral contig discovered in “*P. palustris*” in Indonesia.

## Data Availability

Data supporting reported results are available from GenBank under accession numbers MW503714-28. Total raw data of the 10 RNA libraries are available in Sequence Read Archive (SRA) as BioProject PRJNA788458.

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
