# Peer review of "Natural Populations from the Phytophthora palustris Complex Show a High Diversity and Abundance of ssRNA and dsRNA Viruses"

_jof, 2022, doi:10.3390/jof8111118_

Round 1
Reviewer 1 Report
The paper by Botella et al. reports a high diversity of RNA viruses in samples of Phytophthora palustris complex from its native range in Southeastern and Eastern Asia and defines certain localities as viruses’ hotspots. Furthermore, multiple infections of the same isolate are also confirmed. This study provides insights into the spread and evolution of RNA viruses in natural populations of an oomycete species which has important implication in light of ever more aggressive dissemination of invasive alien tree pathogens.
The paper is very well written, the presentation is well-structured and the topic is relevant in the fields of plant pathology, virology and mycology. The study goals are clearly stated, the methods used are appropriate and the results obtained accomplish the set goals. The discussion is well written, with critical considerations from different angles. The conclusions are sound and consistent with presented results.
The paper is publication-worthy and requires only minor revision. The corrections needed concern mainly wrong references to tables/figures in text, several literature citations missing from reference list and minor language errors. All the corrections and suggestions are marked in comments in attached pdf document.

Author Response
Dear reviewer 1, we have incorporated most of the suggested changes as indicated below and in the word file of the revised manuscript.
- Abstract first line. We have modified the sentence
- Reference number 11.
We have changed the reference for the correct one: Richards TA, Dacks JB, Jenkinson JM et al. Evolution of filamentous plant pathogens: gene exchange across eukaryotic king- doms. Curr. Biol. 2006, 16, 1857–64.
- Lines 54-56: We have changed the sentence in order to be clearer in our statement.
- Line 67: we have added the word text.
- Line 83: we have changed this sentence
- Line 89: change incorporated
- Line 91: change made.
- Line 120: done
- We added the web address of all the online programs used as indicated by the reviewer.
- Line 198: 1X
- Line 211: changed
- Line 212: sentence has been connected.
- Lines 216 and 219: changes suggested has been made and captions reformulated in table S5.
- Line 220: databases
- Table 1: corrected as indicated by the reviewer.
- Lines 232, 242, 244, changes made.
- Suggestion for Table 1: column including the population where the virus was sampled. We appreciate this comment and understand the reviewer’s concern. We are open to do the change, but we believe that the figures 7-9 already contains this information, we invite the reviewer to give us his/her opinion on this matter again.
- Line 248: done
- Line 261. Noted and changed
- Line 302: Since these are not viruses recognized as species by the ICTV, the whole name should remain without italics.
- Line 311. We appreciate this comment very much and made corrections.
- Line 317: corrected
- Line 323: corrected.
- Line 357: change done
- Line 360: change done
- Line 390: corrected
- We apologize for the supplementary table and figure numbering mistakes. All numbers have carefully revised.
- Lines 449 and 450: we appreciate the correction. All the numbers has been checked and corrected.
- Line 472. Corrected
- Line 521: corrected
- Line 546: corrected
- Figure 6. As recommended by the reviewer and trying to fit into the article type-layout given by the journal, figure 6 has been divided into four figures. We hope the results are clearer now. We agree with reviewer and have incorporated the squares to the Japanese and Taiwanese
- Line 624: corrected, it should be Figure S5
- Line 670: corrected
Reviewer 2 Report
The authors present a well-structured paper with a correct methodology and a very detailed results and discussion section. I would like the authors to know that I have selected "major revision" because of the perspective of the study, and not because of the methodology used or the results obtained. The main drawback I see in the article concerns why this study is being done, so I think the "introduction" section should be modified. This paper analyzes the diversity of RNA viruses in 112 isolates of P. palustris from different locations. There is certainly immense merit in the sampling. However, the interest in P. palustris is not adequately justified since it is not described as a pathogenic species complex. Nor is the section on hypovirulence caused by viruses justified. This work is methodologically correct from my point of view, however it is only descriptive, and I would suggest the authors to awaken the reader's interest in this work in the introduction section, perhaps talking more about distribution or evolution of viruses in oomycetes, i.e. the work they have carried out instead of a question of biological control with the hackneyed case of CHV-1, especially when this article deals with a species not catalogued as pathogenic nor the objective is to find viruses that cause hypovirulence. In addition to justifying the study of P. palustris, the authors should justify why they only show interest in RNA viruses and do not analyze DNA viruses that may harbor P. palustris species complex. If the authors have this data, I would advise them to publish it in this paper.
Other minor considerations to be modified:
- Line 43: Only the importance of Phytophthora in forestry system is justified, is it also important in agricultural system?
- Line 138: Please justify why they use this deplection kit, since it is only indicated for human, mouse and rat samples.
- Line 157: The authors should justify why they use the genome of P. parasitica to carry out the methodology. Please, mention closer species to P. palustris and justify that P. parasitica is the closest one which genome is available.
- Not an expert in oomycetes, but could you talk about of mycovirus in case of Phytophthora virus? please, double check it.
- The assignment of identities to viruses is a bit weak. Taking a look at the Identity percentages in Table 1, most are around 50% or less. Is that enough to say that the viruses in the sample belong to the same family as the viruses found in NCBI? It is true that the authors have got very low or next to 0 E-values (that is good, it means that it would be impossible to find random matches between the sequence and a database of the size that the authors have used), but it is still a weak biological signal in my opinion. I would ask the authors provide: 1) to indicate references on identity thresholds in viruses, and/or 2) to select the 10-20 most significant hits in BLASTX and assign the virus to the family that is most frequent (a sort of consensus "family").
- I would say that I have detected some typos in the figure references, please double check them:
o "figures 5a and 5b" should be "figures 6a and 6b" (line 446)
o "figures 4e and 4d" should be "figures 6d and 6e" (line 452)
o "figures 4d and 4e" should be "figures 6d and 6e" (line 454)
o "figures 4b and 4c" should be "figures 6b and 6c" (line 455)
- Finally, I would ask the authors to redo Figure 6. It contains a lot of interesting information but it is a little mess (upside-down, with upside-down legends and totally pixelated). I would suggest splitting it into two figures, one with the maps and the other with the virus and sample tables.
I hope that the authors will receive these comments in a constructive manner.
All the best!
Author Response
Dear reviewer 2, we have incorporated most of the suggested changes as indicated below and in the word file of the revised manuscript.
The authors present a well-structured paper with a correct methodology and a very detailed results and discussion section. I would like the authors to know that I have selected "major revision" because of the perspective of the study, and not because of the methodology used or the results obtained. The main drawback I see in the article concerns why this study is being done, so I think the "introduction" section should be modified. This paper analyzes the diversity of RNA viruses in 112 isolates of P. palustris from different locations. There is certainly immense merit in the sampling. However, the interest in P. palustris is not adequately justified since it is not described as a pathogenic species complex. Nor is the section on hypovirulence caused by viruses justified. This work is methodologically correct from my point of view, however it is only descriptive, and I would suggest the authors to awaken the reader's interest in this work in the introduction section, perhaps talking more about distribution or evolution of viruses in oomycetes, i.e. the work they have carried out instead of a question of biological control with the hackneyed case of CHV-1, especially when this article deals with a species not catalogued as pathogenic nor the objective is to find viruses that cause hypovirulence. In addition to justifying the study of P. palustris, the authors should justify why they only show interest in RNA viruses and do not analyze DNA viruses that may harbor P. palustris species complex. If the authors have this data, I would advise them to publish it in this paper.
We appreciate the reviewer’s point of view and suggestions. Therefore, we have modified, and hopefully, improved the introduction and some parts of the discussion accordingly. On the other hand, although we cannot rule out their existence, we have not detected any DNA virus in our data sets.
Other minor considerations to be modified:
- Line 43: Only the importance of Phytophthora in forestry system is justified, is it also important in agricultural system?
This sentence has been modified.
- Line 138: Please justify why they use this deplection kit, since it is only indicated for human, mouse and rat samples.
Illumina does not offer a specific kit for fungal and oomycete rRNA ribodepletion, and the kit for plants was not recommended by the company. Therefore, for this and previous studies (Botella et al. 2020, Botella et al. 2021 and Raco et al. 2022) we have been using this kit. We agree with the reviewer and also consider that is not the ideal kit but it helps to remove a big portion of unwanted host reads and, consequently, increase the depth of coverage of the resulting viral contigs.
- Line 157: The authors should justify why they use the genome of P. parasitica to carry out the methodology. Please, mention closer species to P. palustris and justify that P. parasitica is the closest one which genome is available.
We chose P. parasitica genome among the available Phytophthora genomes because there was not closely related genome to P. palustris. Although with a closer genome we could have removed a higher portion of host reads, we do not think this can affect the results substantially. Based on our experience (i.e. Raco et al. 2022) there are not significant differences between the final virus sequences detected with or without deleting the host-mapped reads.
- Not an expert in oomycetes, but could you talk about of mycovirus in case of Phytophthora virus? please, double check it.
Yes, for the moment, it is correct. However, as oomycete virus taxonomy is expanding and showing certain differences with fungal viruses, it might change in near future.
- The assignment of identities to viruses is a bit weak. Taking a look at the Identity percentages in Table 1, most are around 50% or less. Is that enough to say that the viruses in the sample belong to the same family as the viruses found in NCBI? It is true that the authors have got very low or next to 0 E-values (that is good, it means that it would be impossible to find random matches between the sequence and a database of the size that the authors have used), but it is still a weak biological signal in my opinion. I would ask the authors provide: 1) to indicate references on identity thresholds in viruses, and/or 2) to select the 10-20 most significant hits in BLASTX and assign the virus to the family that is most frequent (a sort of consensus "family").
The identity percentage shown in Table 1 indicate that the virus sequences found in P. palustris isolates are novel viruses, and potential novel species. The family assignment is not only based on the BLASTX comparison, but it is also based on phylogenetic and genetic analyses. In results and discussion section, we have provided the phylogenetic trees which show how ‘P. palustris’ viruses are related to other viruses, we include the virus regions where the conserved domains are identified together with pfam protein detected. Besides, the alignments of the typical motifs of the different RNA dependent RNA polymerase (RdRp) and nucleoproteins (NC) are included in supplementary data. In addition, viruses are named as “virus-like” (i.e. Phytophthora palustris toti-like virus) not to be too specific and let the International Committee on Taxonomy of Viruses (ICTV) to decide if these viruses correspond to an specific family or genus. For each virus taxon identification, this study has followed the current ICTV rules. We hope the reviewer is satisfied with this explanation.
- I would say that I have detected some typos in the figure references, please double check them:
o "figures 5a and 5b" should be "figures 6a and 6b" (line 446)
o "figures 4e and 4d" should be "figures 6d and 6e" (line 452)
o "figures 4d and 4e" should be "figures 6d and 6e" (line 454)
o "figures 4b and 4c" should be "figures 6b and 6c" (line 455)
We apologize for these mistakes, we have reorganized the order of the figures.
- Finally, I would ask the authors to redo Figure 6. It contains a lot of interesting information but it is a little mess (upside-down, with upside-down legends and totally pixelated). I would suggest splitting it into two figures, one with the maps and the other with the virus and sample tables.
As recommended by the reviewer and trying to fit into the article type-layout given by the journal, figure 6 has been divided into four figures. We hope the results are clearer now.
Round 2
Reviewer 2 Report
We appreciate the reviewer’s point of view and suggestions. Therefore, we have modified, and hopefully, improved the introduction and some parts of the discussion accordingly
Yes, you did it
On the other hand, although we cannot rule out their existence, we have not detected any DNA virus in our data sets.
If you tried to detect, please add that information in a small paragraph including briefly how you did, otherwise justify why you do not search for DNA virus.
Illumina does not offer a specific kit for fungal and oomycete rRNA ribodepletion, and the kit for plants was not recommended by the company. Therefore, for this and previous studies (Botella et al. 2020, Botella et al. 2021 and Raco et al. 2022) we have been using this kit. We agree with the reviewer and also consider that is not the ideal kit but it helps to remove a big portion of unwanted host reads and, consequently, increase the depth of coverage of the resulting viral contigs.
Totally agree. Could you just provide how many reads in % were rRNA with this depletion? I think it is good to point out that Plant researchers also need this technology for good quality results.
The identity percentage shown in Table 1 indicate that the virus sequences found in P. palustris isolates are novel viruses, and potential novel species. The family assignment is not only based on the BLASTX comparison, but it is also based on phylogenetic and genetic analyses. In results and discussion section, we have provided the phylogenetic trees which show how ‘P. palustris’ viruses are related to other viruses, we include the virus regions where the conserved domains are identified together with pfam protein detected. Besides, the alignments of the typical motifs of the different RNA dependent RNA polymerase (RdRp) and nucleoproteins (NC) are included in supplementary data. In addition, viruses are named as “virus-like” (i.e. Phytophthora palustris toti-like virus) not to be too specific and let the International Committee on Taxonomy of Viruses (ICTV) to decide if these viruses correspond to an specific family or genus. For each virus taxon identification, this study has followed the current ICTV rules. We hope the reviewer is satisfied with this explanation.
Great
We chose P. parasitica genome among the available Phytophthora genomes because there was not closely related genome to P. palustris. Although with a closer genome we could have removed a higher portion of host reads, we do not think this can affect the results substantially. Based on our experience (i.e. Raco et al. 2022) there are not significant differences between the final virus sequences detected with or without deleting the host-mapped reads
Agree, I do not think it affects the results, but it affects your efforts, so mention it anywhere (methods or even in the discussion) the closest related genome available to P. palustris. Other researchers should use it preferably.
Author Response
- (A) On the other hand, although we cannot rule out their existence, we have not detected any DNA virus in our data sets.
- (Reviewer) If you tried to detect, please add that information in a small paragraph including briefly how you did, otherwise justify why you do not search for DNA virus.
- We added a sentence to the manuscript to clarify that we did not detect any DNA virus using our data processing pipeline. If there is any, we could have found it because the technique total RNA-Sequencing allows to detect any transcript from any virus (RNA or DNA) genes.
- (A) Illumina does not offer a specific kit for fungal and oomycete rRNA ribodepletion, and the kit for plants was not recommended by the company. Therefore, for this and previous studies (Botella et al. 2020, Botella et al. 2021 and Raco et al. 2022) we have been using this kit. We agree with the reviewer and also consider that is not the ideal kit but it helps to remove a big portion of unwanted host reads and, consequently, increase the depth of coverage of the resulting viral contigs.
- (Reviewer) Totally agree. Could you just provide how many reads in % were rRNA with this depletion? I think it is good to point out that Plant researchers also need this technology for good quality results.
- It would have been interesting to report these data but that the company SeqMe did not provide this information. We decided to use the same kit because in the study about Halophytophthora viruses (Botella et al. 2020) the company (Fasteris) checked the % of rRNA still existing in the sample and it was less than 5%.
- (A) We chose parasiticagenome among the available Phytophthora genomes because there was not closely related genome to P. palustris. Although with a closer genome we could have removed a higher portion of host reads, we do not think this can affect the results substantially. Based on our experience (i.e. Raco et al. 2022) there are not significant differences between the final virus sequences detected with or without deleting the host-mapped reads
- (Reviewer) Agree, I do not think it affects the results, but it affects your efforts, so mention it anywhere (methods or even in the discussion) the closest related genome available to P. palustris. Other researchers should use it preferably.
- As required by the reviewer, we added the following sentence: Since the real host genome is not available, the genome of Phytophthora parasitica was randomly chosen, but two closer genomes became recently available (Phytophthora quininea strain Ex-type BL 54 and Phytophthora macrochlamydospora strain Ex-type BL).